# The Buffer Capacity of Riparian Vegetation to Control Water Quality in Anthropogenic Catchments from a Legally Protected Area: A Critical View over the Brazilian New Forest Code



Carlos Alberto Valera [1,2,3], Teresa Cristina Tarlé Pissarra [2,3], Marcílio Vieira Martins Filho [2,3],
Renato Farias do Valle Júnior [3,4], Caroline Fávaro Oliveira [3,4], João Paulo Moura [5],
Luís Filipe Sanches Fernandes [3,5] and Fernando António Leal Pacheco [3,6,*]

[1] Coordenadoria Regional das Promotorias de Justiça do Meio Ambiente das Bacias dos Rios Paranaíba e Baixo Rio Grande, Rua Coronel Antônio Rios, 951, Uberaba MG 38061-150, Brazil; carlosvalera@mpmg.mp.br

[2] Universidade Estadual Paulista, Faculdade de Ciências Agrárias e Veterinárias, Via de Acesso Prof. Paulo Donato Castellane, s/n, Jaboticabal SP 14884-900, Brazil; teresap1204@gmail.com (T.C.T.P.); marcilio.martins-filho@unesp.br (M.V.M.F.)

[3] POLUS—Grupo de Política de Uso do Solo, Universidade Estadual Paulista (UNESP), Via de Acesso Prof. Paulo Donato Castellane, s/n, Jaboticabal SP 14884-900, Brazil; renato@iftm.edu.br (R.F.d.V.J.); caroline_favaro@hotmail.com (C.F.O.); lfilipe@utad.pt (L.F.S.F.)

[4] Instituto Federal do Triângulo Mineiro, Campus Uberaba, Laboratório de Geoprocessamento, Uberaba MG 38064-790, Brazil

[5] Centro de Investigação e Tecnologias Agroambientais e Biológicas, Universidade de Trás-os-Montes e Alto Douro, Ap. 1013, 5001-801 Vila Real, Portugal; jpmoura@utad.pt

[6] Centro de Química de Vila Real, Universidade de Trás-os-Montes e Alto Douro, Ap. 1013, 5001-801 Vila Real, Portugal

[*] Correspondence: fpacheco@utad.pt; Tel.: +55-351-917519833

**Abstract:** The riparian buffer width on watersheds has been modified over the last decades. The human settlements heavily used and have significantly altered those areas, for farming, urbanization, recreation and other functions. In order to protect freshwater ecosystems, riparian areas have recently assumed world recognition and considered valuable areas for the conservation of nature and biodiversity, protected by forest laws and policies as permanent preservation areas. The objective of this work was to compare parameters from riparian areas related to a natural watercourse less than 10 m wide, for specific purposes in Law No. 4761/65, now revoked and replaced by Law No. 12651/12, known as the New Forest Code. The effects of 15, 30 and 50 m wide riparian forest in water and soil of three headwater catchments used for sugar cane production were analyzed. The catchments are located in the Environmental Protection Area of Uberaba River Basin (state of Minas Gerais, Brazil), legally protected for conservation of water resources and native vegetation. A field survey was carried out in the catchments for verification of land uses, while periodical campaigns were conducted for monthly water sampling and seasonal soil sampling within the studied riparian buffers. The physico-chemical parameters of water were handled by ANOVA (Tukey's mean test) for recognition of differences among catchments, while thematic maps were elaborated in a geographic information system for illustration purposes. The results suggested that the 10, 30 or even 50 m wide riparian buffers are not able to fulfill the environmental function of preserving water resources, and therefore are incapable to ensure the well-being of human populations. Therefore, the limits imposed by the actual Brazilian Forest Code should be enlarged substantially.

**Keywords:** water pollution; riparian forest; environmental Law; anthropogenic catchment; watershed management; land use policy

---

## 1. Introduction

Riparian forests are woodlands in association with streams, rivers and lakes. The location of riparian forests adjacent to water courses ensures that they can exert a strong influence on the quality of freshwater and help to protect the whole ecosystem from anthropogenic activities taking place upwards in the watershed [1–4]. Besides protection, riparian forests provide multiple services such as habitat for aquatic species, soil biodiversity, sediment filtering, flood control, stream channel stability and aquifer recharge [5–13].

The water, soil and vegetation of riparian forests are state indicators of conservation and preservation of land and stream suitability [14]. The biotic community components act as integrator of ecological conditions [12,15,16] and form the transition between the aquatic environment and the anthropogenic pressure. From a different standpoint, the biotic community components express the different spatial and temporal scales of anthropogenic pressures, and therefore support the environmental assessment of watersheds [3,17,18]. For this reason, efforts should be made to understand the theory and metrics of soil attributes and water quality in riparian buffer ecosystems and their link to specific or aggregated types of anthropogenic disturbance [19].

Studies on the width of riparian forests are abundant and relevant [20–22], but only a few works had the main purpose to contribute, from scientific grounds, to the evaluation of environmental laws. In Brazil, riparian forests are called permanent preservation areas (PPA) under the terms of articles 4th, 5th and 6th of Law No. 12651/12 (the so-called New Forest Code), being defined as: "*protected area, covered or not by native vegetation, with the environmental function of preserving water resources, the landscape, the geological stability and the biodiversity, facilitating the gene flow of fauna and flora, protecting the soil and ensuring the well-being of human populations*" [23].

The technical concept of PPA was introduced in the first Brazilian Forest Code, published on the 23rd January 1934 (Federal Decree No. 23793/34), which has categorized the national forest into four types: protected forest, remaining forest, model forest and income forest. Among other roles, the protected forests were meant to preserve the water flow, minimize the erosion process and ensure public health conditions, and therefore fall into the current concept of permanent preservation area. It is worth to mention that the legal concept of PPA, already used in the revoked Federal Law No. 4771/65 and reproduced in the current Federal Law No. 12651/12, was not created by the legislator. Instead, the legislator has appropriated the existing technical and scientific knowledge for the normative definition of that ecosystem. On 1965, when the Forest Law was published, there was no Ministry of Environment, and the environmental terms were themselves incipient, resulting that Federal Law No. 4771/65 was created and managed by the Ministry of Agriculture, Livestock and Supply.

The 1965 and 2012 forest laws were mostly based on the concept of preservation. Other concepts and definitions equally relevant for the role of riparian forests as preservation, such as ecological function or ecosystem service, were not emphasized in these laws. The Ecological function is "*the operation by which the biotic and abiotic elements that are part of a given environment contribute, in their interaction, to the maintenance of the ecological balance and to the sustainability of the evolutionary processes*". By fulfilling this function the PPA would provide ecosystem services through ecological and evolutionary processes, including gene flow, disturbance and nutrient cycling, besides the preservation issue. The ecosystem service concept and the practical assessment of ecosystem services [24] in watersheds [25] should be more explicitly applied to the PPAs of anthropogenic catchments.

The New Forest Code has also reduced the overall protection of riparian forests. The width of riparian buffers has not been altered in the new Law, but the location criteria used to measure it have changed. This has led to a smaller area of protected forests, besides the implications for the

renting of such protected spaces as well as for transition rules (article No. 59 of Federal Law No. 12651/12). The reduction of riparian vegetation reduces the environmental protection of streams provided by these "green filters", and therefore the likelihood of ecological disasters is expected to increase compromising the sustainability of aquatic systems. A coherent forest code should look upon catchments as spaces where man and nature coexist and self-sustain. A different look inevitably opens the space for radically opposing goals based on the same concept [14]. Therefore, a scientifically based assessment of forest laws represents an environmental policy topic worthy of investigation.

This study aims to take that step forward, namely to compare riparian buffer widths as defined by the revoked (Law No. 4771/65; [26]) and current (Law No. 12651/12; [23]) forest laws for the marginal areas of streams less than 10 m wide, and verify their effects on water and soil resources. The specific goals are: (1) to study riparian buffer soils and water quality along watercourses of anthropogenic watersheds, namely watersheds used for sugar cane production. Watercourses in these catchments may be affected by a diversity of pollutants, including nitrogen and phosphorus from fertilizers or fine sediments from soil erosion. In this study, water quality was assessed by an index that involves the measurement of dissolved oxygen, turbidity, total dissolved solids, which means parameters that can be interpreted as proxies to those pollutants. The index is called the *IWQ*—Index for Water Quality and was proposed by the Environmental Company of São Paulo State—CETESB (https://cetesb.sp.gov.br) to be used in water quality assessments. The study was replicated in watercourses with 15, 30 and 50 m wide riparian forests; (2) to look upon the riparian buffer width defined by the New Forest Code and attempt to understand the underlying environmental function of conserving water resources; (3) to define metrics for the evaluation of water and soil resources within riparian buffers. Due to its regional and national importance, this research was carried out in the Uberaba River Basin, namely at the Municipal Environmental Protection Area.

## 2. Materials and Methods

### 2.1. Study Area

The study area comprises the Municipal Environmental Protection Area of Uberaba River Basin (EPA-URB), which is located in the Triângulo Mineiro Region, State of Minas Gerais, Brazil (Figure 1). The EPA-URB occupies an area of approximately 525 km$^2$ between the Meridian coordinates 188–220 km East and Parallel coordinates 7815–7840 km North of Universal Transverse Mercator coordinate system, 23K. The EPA-URB was acknowledged as a Sustainable Land Use Conservation Unit, which is a portion of Minas Gerais State territorial waters subject to a special regime of administration. The demarcation of the EPA-URB involved the recognition of important natural characteristics besides water resources, namely native vegetation (Cerrado Biome), worth of state protection by the Municipal Law No. 9892 of 28 December 2005.

According to Köppen's climate classification, the region is classified as Aw, tropical, and the climatic domain is classified as semi-humid with 4 to 5 dry months, with a relative humidity of 70–75%. The average annual temperature varies between 20 and 24 °C. The warmest months are October to February, with temperatures ranging between 21 and 25 °C. The month of July is the coldest month with temperatures ranging from 16 to 18 °C. The long-term (sixty two year record) mean annual precipitation in Uberaba municipality is 1584.2 mm. On a monthly basis, average rainfall varies between 42.8 and 541 mm (www.inmet.gov.br/).

The EPA-URB is located in the Central Brazil Plateau and northeast portion of Paraná Basin. Topography is characterized by undulated landscapes. Geology is dominated by a sedimentary sequence comprising two major geologic groups and associated formations: the Sao Bento Group and Serra Geral Formation; the Bauru Group and the Marilia and Uberaba formations. The São Bento Group is composed of basalts cropping out at lower altitudes. The Uberaba Formation is made up of Cenozoic sediments, with a predominance of sedimentary rocks with volcaniclastic contribution, and overlays the Serra Geral Formation along an erosive contact. The upper contact with the Marília

Formation is also considered abrupt, being marked by a silexite level and a conglomerate rich in quartz grains cemented by calcite [27].

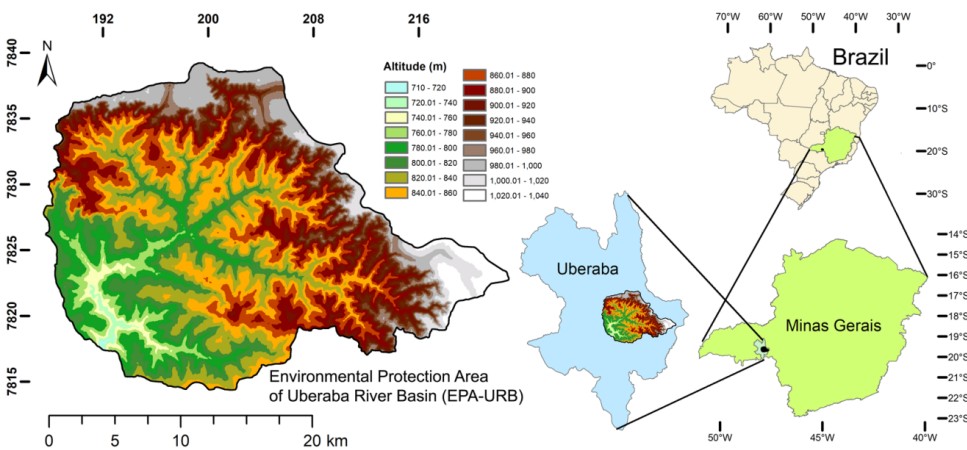

**Figure 1.** Location of the Environmental Protection Area of Uberaba River Basin ( EPA-URB) in the Uberaba Municipality, State of Minas Gerais, and Brazil.

The main soil units are latosols (predominant) and argisols (small areas), according to the Brazilian system of soil classification (https://www.embrapa.br/solos/sibcs). These soil units correspond to ferralsols in the World Reference Base (http://www.fao.org/soils-portal) and oxisols in the classification scheme of Natural Resources Conservation Service (https://www.nrcs.usda.gov/wps/portal/nrcs/site/soils). The latosols are characterized by clayey texture whereas the argisols are characterized by sandy texture.

*2.2. Experimental Sites (Sub-Catchments)*

The experimental sites comprised three sub-basins selected within the EPA-URB, termed Mangabeira 1 (area: 373.09 ha), Mangabeira 2 (426.6 ha) and Lanhoso (1243.64 ha) (Figure 2). In all cases the catchments were mostly used for sugar cane plantations, which occupy 49.4, 39.5 and 34.3% of the area, respectively, and therefore could be considered anthropogenic basins. Besides this use, the catchments were substantially occupied by native forests (36.1, 30.9 and 53.1%). However, the riparian buffers marginal to the watercourses were characterized by quite different widths: on average, 15 m in Mangabeira 1, 30 m in Mangabeira 2 and 50 m in Lanhoso. The samples of soil and water were collected at the sub-basin outlet.

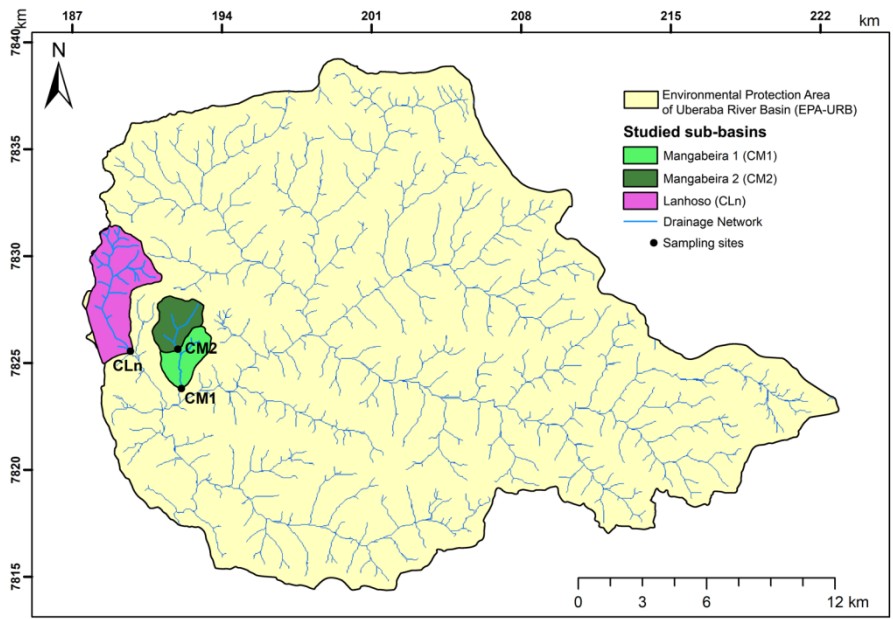

**Figure 2.** Experimental sites (sub-basins Mangabeira 1, Mangabeira 2 and Lanhoso).

Land uses in the three sub-basins are illustrated in Figure 3a–c and can be summarized as follows: *pasture*—natural or managed pastures (used for the grazing of domestic livestock), sometimes composed of grasses and forbs, in other cases including native vegetation; *sugar cane*—sugar cane plantations; *rural dwelling*—space occupied by the people who work on farms and related activities; *native forest*—area occupied by spontaneous native vegetation, sometimes deforested; *managed forest*—mainly eucalyptus stands; *water bodies*—lakes and reservoirs; *roads*—paved roads; *other land uses*—include orchards, and areas used for rain fed or irrigated corps.

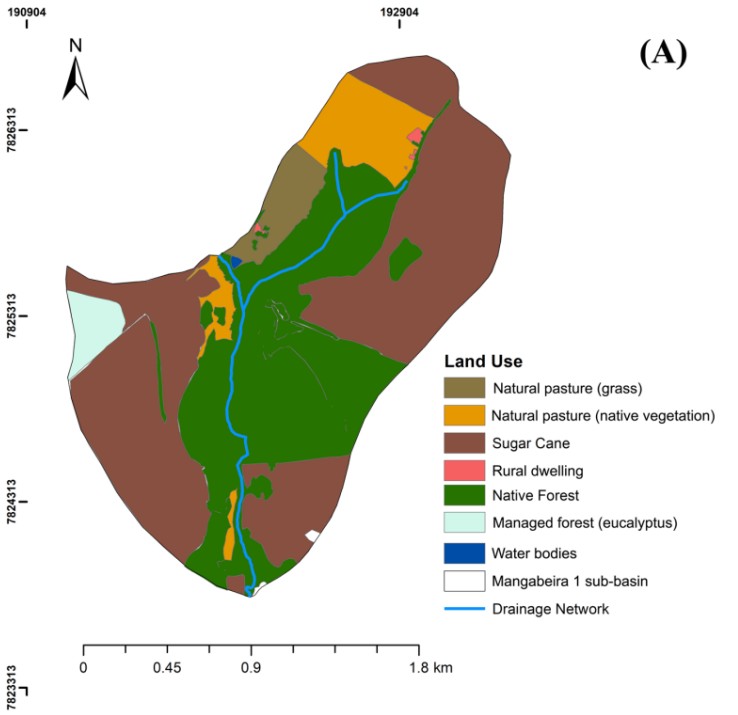

**Figure 3.** *Cont.*

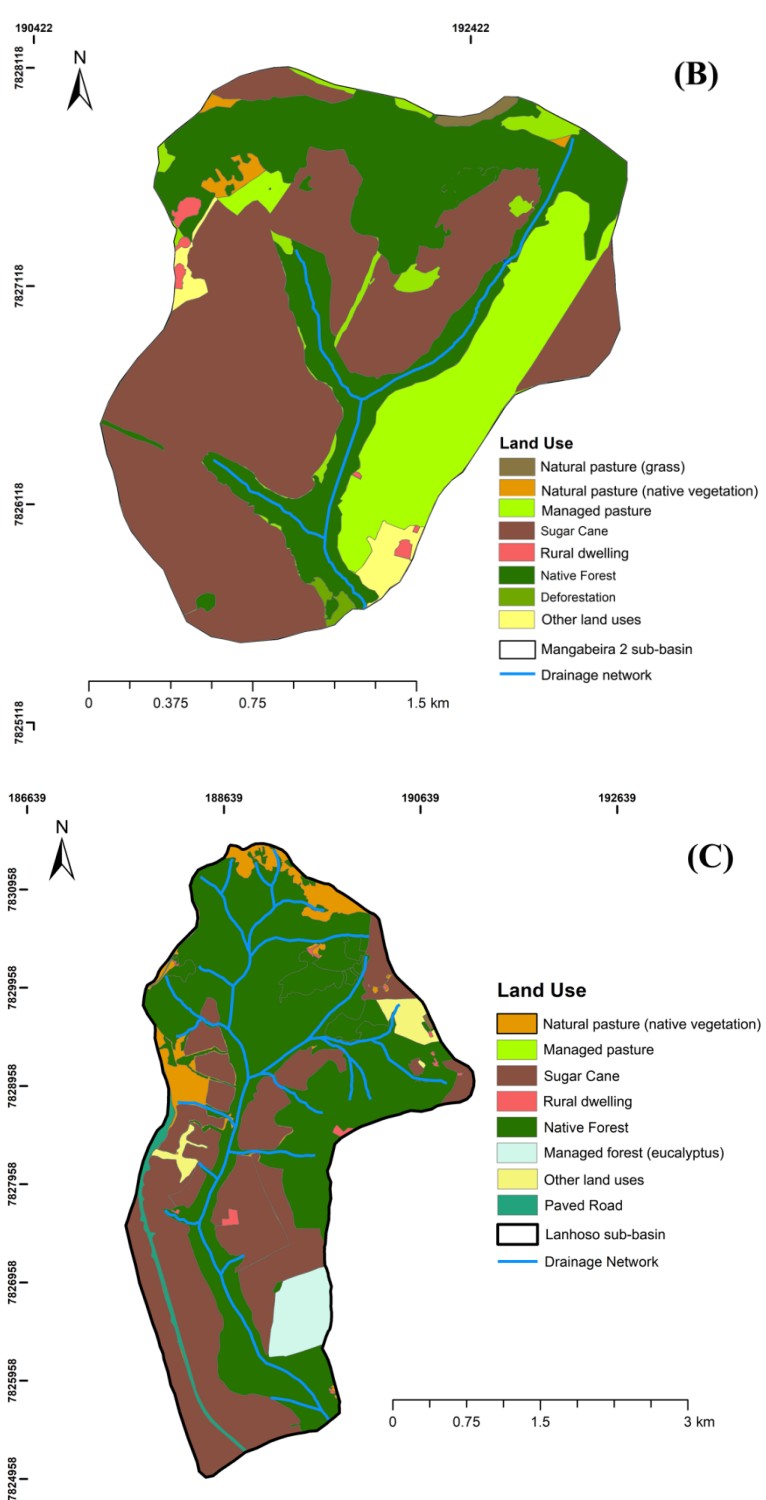

**Figure 3.** (**a**) Land use in Mangabeira 1 sub-basin. Buffer strip width: 15 m; (**b**) Land use in Mangabeira 2 sub-basin. Buffer strip width: 30 m.; (**c**) Land use in Lanhoso sub-basin. Buffer strip width: 50 m.

*2.3. Sampling and Analysis*

2.3.1. Soils

The sampling of soils was carried out in the areas occupied by riparian vegetation. Depending on the sub-basin, this represented a buffer extending 15, 30 or 50 m from the watercourse upwards. The sampling procedure followed the guidelines of São Paulo State Environmental Agency [28],

and took place in April and November of 2015 at least 10 m away from the stream within the buffer zone. The campaigns involved the collection of undisturbed as well as disturbed samples. The undisturbed samples were collected at the 0–20 and 20–40 cm depth layers to determine soil density and other physical attributes. The number of sites per sub-basin was five, and therefore the total number of undisturbed samples was 60 spanning the two sampling seasons. The disturbed soils were collected within a 10 m square grid with 3 columns and 6 rows, in a total of 18 sites (repetitions). Considering the number of seasons (2), the number of sub-basins (3), and the number of sites per sub-basin (18), the amount of disturbed soil samples was 108.

Following collection, the soil samples were air dried, stripped and passed through a 2 mm mesh screen for chemical analyses. The analyses followed the methods of [29] and involved determination of: pH; $Al^{3+}$($cmol_c$ $dm^{-3}$); Ca—exchangeable calcium ($cmol_c$ $dm^{-3}$); Mg—exchangeable magnesium ($cmol_c$ $dm^{-3}$); H+Al—potential acidity ($cmol_c$ $dm^{-3}$); SB—sum of bases ($cmol_c$ $dm^{-3}$); t = SB + $Al^{3+}$—Cation exchange capacity ($cmol_c$ $dm^{-3}$); T—Cation exchange capacity at pH 7.0, calculated as a function of (SB) + (H+Al), expressed as $cmol_c$ $dm^{-3}$; K—exchangeable potassium (mg $dm^{-3}$); P—available phosphorus (mg $dm^{-3}$); V = SB/CEC—Base Saturation (%); m—aluminum saturation (%); SOM—soil organic matter (dag $kg^{-1}$); OC—organic carbon (dag $kg^{-1}$); Sand (%); Silte (%); Clay (%).

### 2.3.2. Water

The stream water samples were collected immediately downstream from the soil sampling sites, in sectors of the stream that were adjacent to the riparian buffer. The sampling took place approximately 60 cm far from the stream margin, every month during 13 months (January 2016–January 2017). The annual rainfall in 2016 was 1214.4 mm. This value is smaller than the long-term average (1584.2 mm), meaning that 2016 was a dry year. Each month, the samples were collected between calendar days 15 and 20. The weather conditions in the sampling day as well as during the three antecedent days are summarized in Table 1. In the sampling day, rainfall was always <5 mm with the exception of February 2016 and January 2017 campaigns, when rainfall reached 5.9 and 10.9 mm, respectively. In the antecedent days, average rainfall was also small (3.5–5.8 mm), with few exceptions represented in bold in Table 1. The antecedent days with a substantial rainfall were 13 November 2016, and 16 January 2017, with precipitation >25 mm. Therefore, the average analytical results should reflect long-term effects of land use and buffer strip width on the quality of stream water rather than short term effects related with storm events. In the sampling site of a catchment, each campaign involved the measurement of water quality parameters in 10 samples (repetitions), according to CONAMA Resolution No. 357/2005. The parameters were measured using a Horiba U-50 Series multi-parameter probe, and comprised: T—water temperature (°C), pH, ORP—Oxidation Reduction Potential (mV), Ec—Electrical conductivity ($\mu$S $cm^{-1}$), Turbidity, measured in Nephelometric Turbidity Units (NTU), DO—Dissolved oxygen (mg $L^{-1}$), PDO—Percentage of Dissolved Oxygen (%), and TDS—total dissolved solids (mg $L^{-1}$).

**Table 1.** Weather conditions (rainfall) in the water sampling day and the three antecedent days (day-1 until day-3). Values larger than 10 mm $day^{-1}$ are represented in boldface.

| | | | | | Water Sampling Date | | | | | | | | |
|---|---|---|---|---|---|---|---|---|---|---|---|---|---|
| Year | | | | | 2016 | | | | | | | | 2017 |
| Month | Jan | Feb | Mar | Apr | May | Jun | Jul | Aug | Sep | Oct | Nov | Dec | Jan |
| Day | 19 | 16 | 15 | 19 | 17 | 21 | 19 | 16 | 20 | 18 | 15 | 20 | 17 |
| | | | | | Rainfall (mm) | | | | | | | | |
| Day | 2.0 | 5.9 | 4.0 | 2.4 | 0.5 | 0.2 | 0.0 | 0.8 | 0.1 | 3.7 | 1.9 | 2.3 | **10.9** |
| Day-1 | 8.0 | 8.7 | 5.5 | 1.2 | 2.8 | 0.0 | 0.0 | 0.0 | 0.7 | 4.7 | **13.9** | 2.6 | **27.0** |
| Day-2 | 3.4 | 2.9 | 2.3 | 0.9 | 0.1 | 0.0 | 0.0 | 0.0 | 0.1 | 4.9 | **32.4** | 5.0 | **10.0** |
| Day-3 | **16.6** | 2.8 | 6.3 | 3.7 | 0.9 | 0.0 | 0.0 | 0.0 | 0.0 | 0.6 | 8.4 | 1.4 | 4.4 |

A subset of parameters was used to calculate the Index for Water Quality (*IWQ*) proposed by the Environmental Company of São Paulo State—CETESB (https://cetesb.sp.gov.br):

$$IWQ = \prod_{i=1}^{n} q_i^{w_i} \tag{1}$$

where $0 \leq IWQ \leq 100$, $q_i$ is the quality of *i*th parameter obtained from standardization of the measured values into a 0–100 range, $w_i$ is the weight of *i*th parameter, which varies in the $0 \leq w_i \leq 1$ interval as function of its importance to the overall quality, and $n$ is the total number of parameters. According to CETESB, $n = 9$ and comprises water temperature, pH, dissolved oxygen, turbidity, total dissolved solids, biochemical oxygen demand, fecal coliforms, total nitrogen and total phosphorus. When data is lacking for some of these parameters, the index can still be calculated using a different set of weights as proposed by [30]. The calculus of *IWQ* in the present study was based on the first five parameters from the list ($n = 5$) and on the following weights: 0.10 (water temperature); 0.21 (pH); 0.17 (turbidity); 0.2 (dissolved oxygen); 0.17 (total dissolved solids). The standardization curves for these parameters, which transform the measured parameters into *q* scores (Equation (1)), are portrayed in Figure 4.

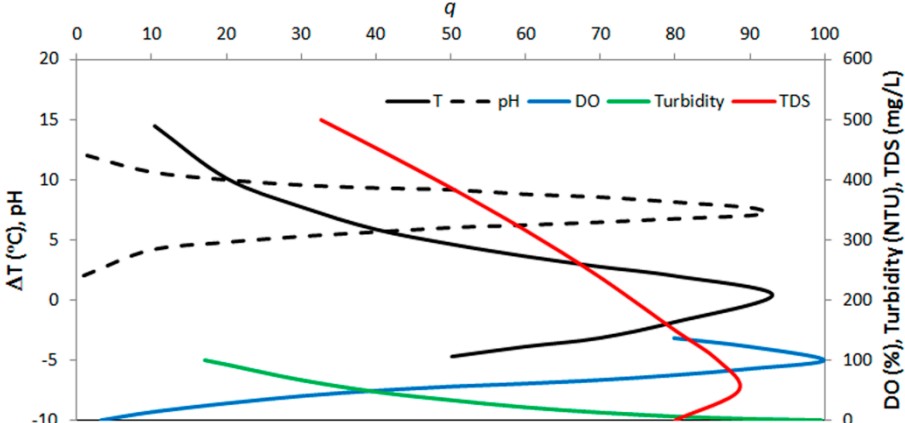

**Figure 4.** Standardization curves used to transform the water quality parameters into *q* scores (Equation (1)). Source: https://cetesb.sp.gov.br.

According to the *IWQ*, the quality of stream water is graded as follows: extremely poor ($IWQ \leq 19$), poor ($19 < IWQ \leq 36$), regular ($36 < IWQ \leq 51$), good ($51 < IWQ \leq 79$), excellent ($79 < IWQ \leq 100$). It is worth to note that the *IWQ* is rather sensitive to small changes in the bearing parameters, given the multiplicative formulation of Equation (1). As corollary of this conception, a good water quality ($IWQ > 51$) requires that all *q* values are high while an excellent quality ($IWQ > 79$) implies that all *q* scores are very high.

### 2.4. Thematic Maps and Statistical Treatment of Soil and Water Data

The thematic maps (e.g., Figures 1–3) were prepared in ArcMap software of ESRI [31], a common tool in spatial analysis of hydrologic and environmental data widely used in many recent studies [32,33]. The base information was compiled from various spatial databases, namely the maps published by the Brazilian Institute for Geography and Statistics (https://ww2.ibge.gov.br) on the 1:100,000 scale, and the digital terrain model obtained from the ASTER GDEN V2 satellite image with spatial resolution of 30 m. The statistical treatment of water data was based on the analysis of variance (ANOVA) and Tukey's mean test ($p < 0.05$). The data were processed in the *R* computer program (https://www.r-project.org/).

## 3. Results

The analytical results are depicted in Table 2 (soils) and Table 3 (water). The water quality index (*IWQ*) is depicted in Table 4.

**Table 2.** Analytical results for the soil samples. The symbols were defined in the text (Section 2.3.1).

| Parameter | pH | Al$^{3+}$ | Ca | Mg | H+Al | SB | t | T | K | P | V | m | SOM | OC | Sand | Silt | Clay |
|---|---|---|---|---|---|---|---|---|---|---|---|---|---|---|---|---|---|
| Unit | | | | | cmolc dm$^{-3}$ | | | | mg dm$^{-3}$ | | % | | dag kg$^{-1}$ | | % | | |
| | | | | | | | | April | | | | | | | | | |
| CM1 | 5.7 | 0.4 | 2.7 | 1.1 | 5.7 | 4.1 | 4.6 | 9.8 | 93.6 | 5.4 | 40.1 | 17.4 | 3.0 | 1.8 | 44.5 | 24.9 | 30.6 |
| CM2 | 5.6 | 0.5 | 1.0 | 0.3 | 4.6 | 1.4 | 1.9 | 6.0 | 41.5 | 9.8 | 25.9 | 32.0 | 4.9 | 2.9 | 63.7 | 23.8 | 12.5 |
| Ln | 5.8 | 0.6 | 2.2 | 0.9 | 4.5 | 3.2 | 3.8 | 7.7 | 54.6 | 2.8 | 43.4 | 17.3 | 3.1 | 1.8 | 60.5 | 23.4 | 16.1 |
| | | | | | | | | November | | | | | | | | | |
| CM1 | 5.7 | 0.4 | 2.7 | 1.1 | 5.7 | 4.1 | 4.6 | 9.8 | 93.6 | 5.4 | 40.1 | 17.4 | 3.0 | 1.8 | 44.5 | 24.9 | 30.6 |
| CM2 | 5.6 | 0.5 | 1.0 | 0.3 | 4.6 | 1.4 | 1.9 | 6.0 | 41.5 | 9.8 | 25.9 | 32.0 | 4.9 | 2.9 | 63.7 | 23.8 | 12.5 |
| CLn | 5.8 | 0.6 | 2.2 | 0.9 | 4.5 | 3.2 | 3.8 | 7.7 | 54.6 | 2.8 | 43.4 | 17.3 | 3.1 | 1.8 | 60.5 | 23.4 | 16.1 |

**Table 3.** Analytical results for the water samples: average value, standard deviation, Tukey's mean test result (ANOVA; $p < 0.05$). The symbols were defined in the text (Section 2.3.2). Values with different label (lowercase letters *a*, *b*, *c* or *d*) are considered significantly different from each other by the Tukey's test, and therefore can be used to differentiate the sub-basins.

| Sub-Basin | Mangabeira 1 | Mangabeira 2 | Lanhoso |
|---|---|---|---|
| | CM1 (15 m) | CM2 (30 m) | CLn (50 m) |
| T (°C) | 19.04 | 20.08 | 19.67 |
| | ±2.6 | ±2.3 | ±2.6 |
| | *d* | *bc* | *cd* |
| pH | 7.10 | 7.00 | 7.43 |
| | ±0.5 | ±0.5 | ±0.4 |
| | *b* | *b* | *a* |
| ORP (mV) | 140.12 | 153.24 | 204.68 |
| | ±34.9 | ±43.2 | ±74.6 |
| | *cd* | *bc* | *a* |
| Ec (µS/cm) | 60 | 70 | 90 |
| | ±20 | ±20 | ±30 |
| | *bc* | *b* | *a* |
| Turbidity (NTU) | 6.25 | 3.75 | 2.03 |
| | ±6.1 | ±2.9 | ±2.0 |
| | *cd* | *d* | *d* |
| DO (mg/L) | 7.64 | 7.32 | 11.58 |
| | ±1.3 | ±1.6 | ±11.3 |
| | *b* | *b* | *a* |
| PDO (%) | 84.72 | 83.23 | 124.81 |
| | ±14.4 | ±18.7 | ±111.01 |
| | *b* | *b* | *a* |
| TDS (mg/L) | 40 | 50 | 60 |
| | ±10 | ±11 | ±19 |
| | *bc* | *b* | *a* |

**Table 4.** Water quality index (*IWQ*) and potentially related environmental variables.

| Sub-Basin | Buffer Width (m) | Sugar Cane—SC (%) | Native Forest—NF (%) | NF/SC | IWQ | Water Quality |
|---|---|---|---|---|---|---|
| CM1 | 15 | 49.4 | 36.1 | 0.7 | 30.8 | Poor |
| CM2 | 30 | 39.5 | 30.9 | 0.8 | 31.0 | Poor |
| CLn | 50 | 34.2 | 53.1 | 1.6 | 33.4 | Poor |

The relationship between soil parameters and the riparian buffer width (Table 2) was not detected for most parameters. However, the percentage of sand and the content of aluminum increased as function of width, while the silt content decreased. The results for sand and aluminum seem to expose the capacity of buffer strips to retain mineral aggregates, especially the more coarse grained. The results for silt may be apparent because sand, silt and clay in a texture analysis sum 100% and therefore when a fraction increases the other tend to decrease regardless of their abundance in the sample.

The analytical results for water (Table 3) indicate a statistical difference between the Mangabeira catchments (CM1 and CM2) and the Lanhoso catchment (CLn), in the case of pH, oxidation-reduction potential, conductivity, dissolved oxygen (in mg L$^{-1}$ or %) and total dissolved solids, which means the majority of parameters. This is strong indication that the catchment with a wider riparian buffer is different from the catchments with a narrower buffer, as regards water quality. The results of the *IWQ* calculation showed reduced values in all basins. These results qualified the stream waters as poor.

When the *IWQ* parameter was plotted as a function of buffer width (Figure 5a) and a trend line was fitted to the scatter points, the fitting equation was parabolic:

$$IWQ = 0.003\,BW^2 - 0.1238BW + 31.971 \tag{2}$$

where *BW* means buffer width. A similar plot, but of *IWQ* as function of *BW* combined with land use/occupation (*NF/SC* = native forest/sugar cane ratio; Figure 5b), could be fitted to the following linear equation:

$$IWQ = 0.0391BW\frac{NF}{SC} + 30.26 \tag{3}$$

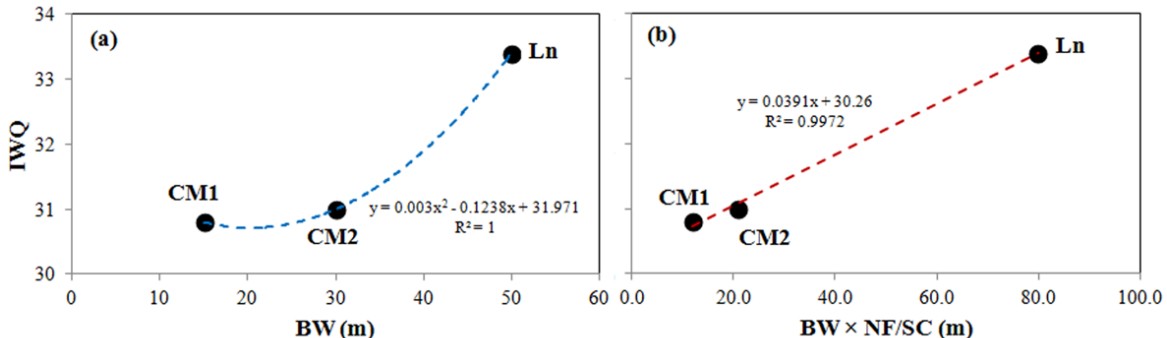

**Figure 5.** Plot of water quality index (*IWQ*) as function of: (**a**) riparian buffer width (*BW*); (**b**) buffer strip width combined with native forest/sugar cane ratio (*BW* × *NF/SC*). The points represent the values of *IWQ* versus *BW* or *IWQ versus BW* × *NF/SC* in the Mangabeira 1 (CM1), Mangabeira 2 (CM2) and Lanhoso (CLn) sub-basins.

Equation (2) attempts to describe the independent influence of buffer strip width on water quality, while Equation (3) analyzes this influence but coupled the potential interference of land use. In fact, the two effects are barely separable because an increase of buffer strip width tends to increase the NF/SC ratio.

The parabolic trend in Figure 5a may reflect the fact that water quality in CM1 is affected by water quality of CM2, besides the influence of local land use and buffer strip width. Because the *IWQ* in both catchments are similar, regardless the differences between buffer strip widths, the contribution from CM2 is probably large. Put another way, if water quality in CM1 reflected solely land use and buffer strip width, then a lower *IWQ* would be expected in this catchment. In the figure, the CM1 point would drop along the *y*-axis and the general trend would shift from the parabolic trend towards a (more likely) linear trend.

Using Equation (3) with constant *NF/SC* values allows relating water quality (*IWQ*) and buffer strip width for specific land occupations. Table 5 describes these relationships for three *NF/SC* ratios: 0.7 (the lowest ratio in the studied catchments), 1.6 (highest ratio) and 3.2 (twice the highest ratio, forecasting implementation of conservation practices through expansion of native forest). In the first case (*NF/SC* = 0.7), it is expected that a regular water quality is attained for *BW* = 205 m, while this value reduced to *BW* = 90 m for *NF/SC* = 1.6, and to *BW* = 45 m for *NF/SC* = 3.2. If the buffer strip width would duplicate in the Lanhoso catchment through implementation of conservation practices, the water quality would become good for *BW* = 155 m. Therefore, the 30 m threshold foreseen in Federal Law No. 12651/2012 may not satisfy the water quality requirements in the studied basins, while it is worth remembering that these catchments are located in a protected area for water resources. It should be admitted, however, that this study was based on a small number of catchments and that more general conclusions would require a more exhaustive analysis based on a larger sample. Besides, the interpretations so far hold for anthropogenic catchments used for sugar cane production and cannot be extended directly to other land uses. Finally, the weather reference for this study is a dry year.

**Table 5.** Riparian buffer width (*BW*), native forest/sugar cane ratio (*NF/SC*) and related water quality (*IWQ*), as predicted by Equation (3) for three constant values of *NF/SC*.

| Water Quality | IWQ | Riparian Buffer Width—*BW* (m) | | |
|:---:|:---:|:---:|:---:|:---:|
| | | *NF/SC* = 0.7 | *NF/SC* = 1.6 | *NF/SC* = 3.2 |
| Poor | 19–36 | 15–205 | 15–90 | 15–45 |
| Regular | 36–51 | 205–300 | 90–300 | 45–155 |
| Good | 51–79 | nd | nd | 155–300 |
| Excellent | 79–100 | nd | nd | nd |

It is also worth recalling that water quality expose the aggregate effects of all natural processes and anthropogenic inputs [34,35] that can occur along the flow paths [36], namely chemical weathering [37–40], uptake/release from/to biota [41], leachates from fertilizers that also affect chemical weathering [42–44], discharge of domestic sewage [45,46], among others. The trend depicted in Figure 5b exposes the impact of sugar cane on *IWQ* combined with the buffering capacity of native forest. It is expected that in absence of native forest (*NF* = 0) the water quality index would drop to *IWQ* = 30.26 because *NF* = 0 implies *BW* × *NF/SC* = 0. This rather low level of *IWQ* would represent the impact on water quality exclusively attributed to sugar cane production (*IWQ*$_0$), including the effects related to fertilizing and management (e.g., erosion control). According to Equation (3), if fertilizing and management conditions are kept unaltered in the future the regular water quality can only be achieved if the proportion of native forest over sugar cane plantations rises substantially. This may not be economically feasible, the reason why the route to follow is to improve management practices to raise the value of *IWQ*$_0$.

## 4. Discussion

The Environmental Protection Area of Uberaba River Basin (EPA-URB) and other similar conservation units exist to reconcile human occupation with the sustainable use of their natural

resources, not to expel human populations. However, the activities and uses developed in these areas are subject to specific rules. This work exposed the need to manage properly the permanent preservation areas of three small catchments located in the EPA-URB to accomplish environmental sustainability.

The capacity of riparian buffers to retain particles and dissolved compounds from catchment uplands depend on the buffer width (50, 30 and 15 m) as well as on the types of land use or occupation and their management practices. This study exposed a stronger buffer capacity in the catchment where the riparian forests extend 50 m upwards from the stream margin, but even in this catchment the water quality is generally poor. A regular or good quality would require much wider strips and larger *NF/SC* ratios. In complement, better management practices could be implemented to prevent or at least reduce substantially the exports of sediment and nutrients towards the streams.

The Brazilian law defined the buffer width limits based on two scenarios: the Federal Law No. 4771 and the New Forest Code. In the first case, for watercourses up to 10 m wide the permanent preservation areas need to extend at least 30 m upwards from the stream margin considering the widest seasonal riverbank. In the second case, there are two rules: The transition rule takes into account the size of land property calculated as fiscal modules and creates a distance from the stream margin that goes from a minimum of 5 m to a maximum of 20 m, considering the regular river bank; the permanent rule defines 30 m as unique distance. By changing the reference from the largest riverbank (wet season) to the regular riverbank, the New Forest Code has decreased the legal riparian buffer width. A huge amount of scientific literature has reported the importance of riparian buffer width for water quality and ecological functions [3,11–14,18]. In the present study, it was suggested for a range of native forest/sugar cane occupational ratios (0.7–3) that the legal width should be at least 45 m, but preferably more, corroborating the studies of GAEMA [47]. Besides, efforts should be made to better understand the theory and the metrics of soil attributes and water quality in riparian forest ecosystems to develop ecological functions for these areas based on buffer width [19].

In general, the owner of a rural property has the legal right to use, enjoy, possess and dispose of it. However, this legal right is not applicable to permanent preservation areas included in the property. The permanent preservation area is a legal area. It is not an area for the socioeconomic use of a land owner. The permanent preservation areas are subject to a restriction of use imposed by the Brazilian Constitution, which aims to ensure a provisional ecosystem function, namely the provision of soil and water as resource. In this context, the management of permanent preservation areas is allowed solely when there is no local option of public and social interest, and the interventions are to be done with a low impact. The New Forest Code recommends a riparian buffer width that can keep fundamental ecosystem functions. This study suggested that this width should be increased to 45 m, at least. Besides the correct dimensioning of the riparian buffer width, a number of mitigation measures are ought of implementation to increase the $IWQ_0$ far above its current value ($IWQ_0 = 30.26$ = poor water quality). To become effective, the causes and paths of pollution should be assessed [48–51] and then the measures should be modeled in spatial decision support systems focused on water resources planning and management [52–56], evidenced and discussed by government agencies and public and private companies, and integrate public policies and environmental management plans [57]. From a broader standpoint, the specific widths along the drainage network should be reviewed, being defined as function of basin area, watercourse and the catchments' social and economic importance for public water supply.

The results illustrated in Figure 5b raised a striking question: What should be the area released to the agro system, in replacement of forest areas that have the aim of protecting the environment? The criticism we make to the New Forest Code in this case, is that the ratio of permanent preservation area over area used for agriculture or other anthropogenic activities, should be defined technically and not on the basis of political or socioeconomic convenience. This rationale is also valid for riparian buffer widths. The technical work of Kageyama, Cordeiro and Metzer [47] strongly suggested a minimum buffer strip width of 50 m to protect small streams from anthropogenic activities located

upward of the catchment hillsides. In the present study, the data collected on water are in favor of a buffer strip width even larger than that threshold if good quality water is aimed at the studied catchments. If anthropogenic activities are practiced in this protected area, the erosion and transport processes, followed by the silting and eutrophication of stream water, will be accentuated, a situation that will be defined as environmental damage according to Art. No. 3 of the Federal Law No. 6938/81 and handled through the "polluter-pays" principle [58].

The present study is corroborated in the literature, and was founded on principles of Environmental Law, namely the principles of precaution and prevention. The New Forest Code should take this and other scientific studies as example, and always be interpreted as *pro-nature*: in favor of Environment. Thus, if field data suggest that environmental vulnerability occurs within 50 m from the stream margin at the widest riverbank, it is clear that one should opt for riparian buffer solutions that result in greater environmental protection.

## 5. Conclusions

The role of riparian vegetation and forest cover in the control of stream water quality in anthropogenic catchments was investigated in this study. The analysis involved three headwater catchments characterized by increasing buffer strip widths, namely 15, 30 and 50 m widths, as well as increasing native forest to sugar cane ratios ($NF/SC$). The studied basins are located in the Environmental Protection Area of Uberaba River Basin (EPA-URB; state of Minas Gerais, Brazil). The water quality analysis aimed to evaluate a recent forest law (Law No. 12651/12) in this very important water resources and native forest (Cerrado Biome) conservation unit. A linear trend was defined between a specific water quality index ($IWQ$) and the combined protective effects of buffer with ($BW$) and $NF/SC$ ($BW \times NF/SC$). Presently, the quality of stream water in the three catchments is poor ($IWQ < 36\%$). The linear trend allows estimating a regular water quality ($36 \leq IWQ \leq 51\%$) if buffer widths were larger than 45 m, but only if the coverage by native forest increased substantially (e.g., duplicated) in the studied basins. Under the current land use ($0.7 \leq NF/SC \leq 1.6$) the regular water quality would be reached for buffer strip widths in the 90–205 m range. While keeping the current $BW$ an $NF/SC$ values, water quality could be improved if conservation practices were implemented in the sugar cane fields to reduce the export of sediments and nutrients towards the aquatic media. Overall, it was suggested in this study that the 30 m buffer strip width proposed in the New Forest Code, is barely capable of protecting water quality in the EPA-URB.

**Author Contributions:** Conceptualization, C.A.V. and T.C.T.P.; methodology, C.A.V. and T.C.T.P.; software, R.F.d.V.J. and J.P.M.; validation, C.A.V., L.F.S.F. and F.A.L.P.; formal analysis, C.A.V., F.A.L.P., T.C.T.P. and M.V.M.F.; investigation, C.A.V.; resources, T.C.T.P.; data curation, C.A.V., C.F.O. and T.C.T.P.; writing—original draft preparation, C.A.V.; writing—review and editing, F.A.L.P.; visualization, R.F.d.V.J.; supervision, T.C.T.P. and M.V.M.F.; project administration, T.C.T.P. and M.V.M.F.; funding acquisition, T.C.T.P., M.V.M.F. and R.F.d.V.J.

**Funding:** The present study was carried out within the framework of the Post Graduation Research Programme of Coordenação de Aperfeiçoamento de Pessoal de Nível Superior (CAPES); Conselho Nacional de Desenvolvimento Científico e Tecnológico (CNPq); Agência do Ministério da Ciência, Tecnologia, Inovações e Comunicações (MCTIC); and Land Use Policy Brazilian Group (POLUS). The author is affiliated with IFTM Renato Farias do Valle Júnior wishes to acknowledge the funding through the CNPq research scholarship Proc. 307921/2018-2. The authors integrated in the CITAB research center were financed by National Funds of the FCT–Portuguese Foundation for Science and Technology POCI-01-0145-FEDER-006958, under the project UID/AGR/04033/2019. The author integrated in the CQVR was funded by National Funds of the FCT–Portuguese Foundation for Science and Technology POCI-01-0145-FEDER-006958, under the project UID/QUI/00616/2019.

**Acknowledgments:** Hygor Evangelista Siqueira, Mauro Ferreira Machado and Renata Cristina Araújo Costa are acknowledged for fruitful discussions, sharing of information and mapping of the area.

**Conflicts of Interest:** The authors declare no conflict of interest. The funders had no role in the design of the study; in the collection, analyses, or interpretation of data; in the writing of the manuscript, or in the decision to publish the results.

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
