# Peer review of "The Buffer Capacity of Riparian Vegetation to Control Water Quality in Anthropogenic Catchments from a Legally Protected Area: A Critical View over the Brazilian New Forest Code"

_water, doi:10.3390/w11030549_

Round 1

Reviewer 1 Report

The article presents the results of a research carried out in an agricultural area of Brazil characterised by the cultivation of Sugar Cane. The authors want to demostrate the importance of the conservation of the riparian buffer in guaranteeing a sufficient filter activity protecting the stream water from pollutants. Results show that where the riparian buffer is wider, the water quality is higher; moreover, they show how the ratio between Sugar Cane and Native Forest extensions also affects the water quality. Basing on their findings, the authors propose two equations that allow to predict the water quality (IWQ) on the base of these two parameters.

The article is well written and reads easily, but it is affected by some important limitations. The main one is that the two main drivers of water quality (Buffer width and NF/SC ratio) are treated separately, but the authors do not show a sufficient dataset that could be used as a base for a statistical separation of the two effects (the experimental basins are just three). Consequently, they lack in consistency in their conclusions, in my opinion not sufficiently proved by data.

To avoid the rejection of the paper, I suggest the authors to discuss and conclude their significant experimental work evidencing its limits and avoiding conclusions which are not sufficiently based on data. Here below some specific comments are reported.

Major comments

1)      Lines 79-99 I am not sure that this is the appropriate register for a scientific paper. I mean, you are attacking the contents of the new law before that you quantify which are the effects of it. I agree that there might be negative consequences, and I undestand the concern of the authors, but I would not argue it that clearly without showing that this is proven by data. I find that the authors should keep a higher position, emphasizing the importance of riparian buffers that the scientific community has already widely demostrated, and hypothesizing the negative consequences that the new law could have by citing references about case studies where this has been already demostrated in a striking way. I am referring particularly to “strong” sentences such as :

LINE 89: “to expose how inefficient the New Forest Code can be”

LINE 98: “which is not the case of Federal Law n.º 12651/12”

2)       It s not clear throughout the paper which are the pollutants produced by the cane sugar cultivation. Nitrogen and phosphorus from fertilizers? Fine sediments from soil erosion? Organic matter? Pesticides? If this is not stated clearly in the introduction, it is not clear to the reader what is the negative impact of anthropogenic activities that the riparian buffer can mitigate.

3)       Lines 235-236 how do you explain this?

4)       Water quality in CM1 is affected also by the water quality of CM2, not only by the land use and riparian buffer width of the river stretch that runs along its main stream: the authors do not deal with it sufficiently.

5)       Equations 1a, 1b and 2; and Figure 4: the regression laws are affected by a huge weakness: a) the authors are using a regression law of IWQ in function of Buffer width, but this is not taking into account NF/SC; while b) the authors are using a regression law of IWQ in function of NF/SC, but this is not taking into account the effect of Buffer width;

Since it is not possible to separate the two effects, the authors should propose a coefficient able to take into account both the factors together, or propose a normalization of the buffer effect respect to the NF/SC ratio. Basing on the presented data, they cannot propose further dissertation on the single effect of just one of the factors (i.e. lines 275-280). Moreover, a regression law based on just 3 points is hard to be considered a robust and reliable trend.

6)       Any discussion about the observed water quality data should consider the combination of NF/SC and Buffer width (for example, as it is reported in lines 289-291) rather than the two factors separated.

7)       Lines 307-309: this is not true. What the authors demonstrated is valuable for a specific NF/SC ratio? . What if there is a different NF/SC ratio? Or rather a different crop?

MINOR COMMENTS

Figure 3: please add letters (3a, 3b, 3c) for more clarity. The Legend about land use is not totally consistent with lines 158-166. For example “Floodplain” is not reported in the figures, and should be distinguished at least in sub categories such as “vegetated flood plain with threes, Vegetated flood plain with grass, non-vegetated floodplain”. Or just eliminate it including those areas in other land use classes

Lines 297-305: I suggest to move it to the introduction section, where the laws are presented

From which reference point did the authors measure the buffer width? Widest riverbank, or ordinary? Could you (or does the law) provide a return period to detect the reference width of the riverbed?

Author Response

Reviewer general appreciation

The article presents the results of a research carried out in an agricultural area of Brazil characterised by the cultivation of Sugar Cane. The authors want to demostrate the importance of the conservation of the riparian buffer in guaranteeing a sufficient filter activity protecting the stream water from pollutants. Results show that where the riparian buffer is wider, the water quality is higher; moreover, they show how the ratio between Sugar Cane and Native Forest extensions also affects the water quality. Basing on their findings, the authors propose two equations that allow to predict the water quality (IWQ) on the base of these two parameters.

The article is well written and reads easily, but it is affected by some important limitations. The main one is that the two main drivers of water quality (Buffer width and NF/SC ratio) are treated separately, but the authors do not show a sufficient dataset that could be used as a base for a statistical separation of the two effects (the experimental basins are just three). Consequently, they lack in consistency in their conclusions, in my opinion not sufficiently proved by data.

Author's answer

We very much appreciate the time and effort put by the reviewer in reading and commenting this manuscript. All the comments were welcome and they were all addressed thoroughly. We did the utmost to comply with the reviewer suggestions. In fact the number of catchments is small and therefore the results need to be written in that context. We emphasized this limitation and rewritten some sentences to alert for the provisional character of our results.

Reviewer general comment.

To avoid the rejection of the paper, I suggest the authors to discuss and conclude their significant experimental work evidencing its limits and avoiding conclusions which are not sufficiently based on data. Here below some specific comments are reported

Authors response:

We agree with the comment. We included sentences to clarify that the dataset has some limitations. For example, before Table 4, where the water quality index is presented for various buffer strip widths considering various NF/SC ratios, the following sentence was added:

“It should be admitted, however, that this study was based on a small number of catchments and that more general conclusions would require a more exhaustive analysis based on a larger sample. Besides, the interpretations so far hold for anthropogenic catchments used for sugar cane production and cannot be extended directly to other land uses. Finally, the weather reference for this study is a dry year.

Reviewer Major comments:

Comment #1

1)      Lines  79-99 I am not sure that this is the appropriate register for a  scientific paper. I mean, you are attacking the contents of the new law  before that you quantify which are the effects of it. I agree that there  might be negative consequences, and I undestand the concern of the  authors, but I would not argue it that clearly without showing that this  is proven by data. I find that the authors should keep a higher  position, emphasizing the importance of riparian buffers that the  scientific community has already widely demostrated, and hypothesizing  the negative consequences that the new law could have by citing  references about case studies where this has been already demostrated in  a striking way. I am referring particularly to “strong” sentences such  as:

LINE 89: “to expose how inefficient the New Forest Code can be”

LINE 98: “which is not the case of Federal Law n.º 12651/12”

Authors response:

In lines 89 and 98 we removed those parts of the sentences.

We also modified the reminder of the paragraph:

“The 1965 and 2012 forest laws were based on the concept of preservation. In both cases, but especially in the case of Federal Law n.º 12651/12 the legislator was not able to capture from modern science new concepts and definitions, as relevant for the role of riparian forests as preservation, which are for example the concepts of ecological function or ecosystem service. The Ecological function is “the operation by which the biotic and abiotic elements that are part of a given environment contribute, in their interaction, to the maintenance of the ecological balance and to the sustainability of the evolutionary processes". By fulfilling this function the PPA would provide ecosystem services through ecological and evolutionary processes, including gene flow, disturbance and nutrient cycling, besides the preservation issue. The ecosystem service concept and the practical assessment of ecosystem services [24] in watersheds [25] must therefore be applied to the PPAs of anthropogenic catchments.”

which is now written as:

“The 1965 and 2012 forest laws were mostly based on the concept of preservation. Other concepts and definitions equally relevant for the role of riparian forests as preservation, such as ecological function or ecosystem service, were not emphasized in these laws. The Ecological function is “the operation by which the biotic and abiotic elements that are part of a given environment contribute, in their interaction, to the maintenance of the ecological balance and to the sustainability of the evolutionary processes". By fulfilling this function the PPA would provide ecosystem services through ecological and evolutionary processes, including gene flow, disturbance and nutrient cycling, besides the preservation issue. The ecosystem service concept and the practical assessment of ecosystem services [24] in watersheds [25] should be more explicitly applied to the PPAs of anthropogenic catchments.

We also modified the sentence:

“In addition to ignoring the safeguard of ecosystem services, the New Forest Code has reduced the overall protection of riparian forests.”

which is now written as

“The New Forest Code has also reduced the overall protection of riparian forests.”

Comment #2

2)       It s not clear throughout the paper which are the pollutants produced by the cane sugar cultivation. Nitrogen and phosphorus from fertilizers? Fine sediments from soil erosion? Organic matter? Pesticides? If this is not stated clearly in the introduction, it is not clear to the reader what is the negative impact of anthropogenic activities that the riparian buffer can mitigate.

Authors response:

We thank the reviewer for this pertinent comment. In the revised version we changed the scope of objective #1 as follows.

Where it was

“1) to study riparian buffer soils and water quality along watercourses of anthropogenic watersheds.”

Now is:

“1) to study riparian buffer soils and water quality along watercourses of anthropogenic watersheds, namely watersheds used for sugar cane production. Watercourses in these catchments may be affected by a diversity of pollutants, including nitrogen and phosphorus from fertilizers or fine sediments from soil erosion. In this study, water quality was assessed by an index that involves the measurement of dissolved oxygen, turbidity, total dissolved solids, which means parameters that can be interpreted as proxies to those pollutants. The index is called IWQ – Index for Water Quality and was proposed by the Environmental Company of São Paulo State – CETESB (https://cetesb.sp.gov.br) to be used in water quality assessments.

Comment #3

3)       Lines 235-236 how do you explain this?

Authors response:

The following justification was added to the revised version:

“The results for sand and aluminum seem to expose the capacity of buffer strips to retain mineral aggregates, especially the more coarse grained. The results for silt may be apparent because sand, silt and clay in a texture analysis sum 100% and therefore when a fraction increases the other tend to decrease regardless of their abundance in the sample.”

Comment #4

4)       Water quality in CM1 is affected also by the water quality of CM2, not only by the land use and riparian buffer width of the river stretch that runs along its main stream: the authors do not deal with it sufficiently.

Authors response:

This is a very pertinent point. In the revised version we addressed it by adding the following sentence to the results section:

“The parabolic trend in Figure 4a may reflect the fact that water quality in MG1 is affected by water quality of MG2, besides the influence of land use and buffer strip width. Because the IWQ in both catchments are similar, regardless the differences between buffer strip width, the contribution from MG2 is probably large. Put another way, if water quality in MG1 reflected solely land use and buffer strip width, then a lower IWQ would be expected in this catchment. In the figure the MG1 point would drop along the y-axis and the general trend would shift from the parabolic trend towards a more expected linear trend.”

Comment #5

5)       Equations 1a, 1b and 2; and Figure 4: the regression laws are affected by a huge weakness: a) the authors are using a regression law of IWQ in function of Buffer width, but this is not taking into account NF/SC; while b) the authors are using a regression law of IWQ in function of NF/SC, but this is not taking into account the effect of Buffer width;

Since it is not possible to separate the two effects, the authors should propose a coefficient able to take into account both the factors together, or propose a normalization of the buffer effect respect to the NF/SC ratio. Basing on the presented data, they cannot propose further dissertation on the single effect of just one of the factors (i.e. lines 275-280). Moreover, a regression law based on just 3 points is hard to be considered a robust and reliable trend.

Authors response:

The reviewer is totally right. In a multiple regression involving IWQ, BW and NF/SC we would expect the active role of independent terms (influenced solely by BW or NF/SC) as well as of interference terms (influenced by the product BW × NF/SC). In the revised version we kept Figure 4a unchanged to represent the term related with BW, but changed Figure 4b to describe the combined influence of BW and NF/SC on IWQ. Then, we discussed the consequences of BW for IWQ considering various NF/SC values (revised Table 4)

Comment #6

6)       Any discussion about the observed water quality data should consider the combination of NF/SC and Buffer width (for example, as it is reported in lines 289-291) rather than the two factors separated.

Authors response:

We agree. The revised Figure 4b and revised Table 4, as well as the concomitant discussion, are based on a combined analysis of BW and NF/SC.

Comment #7

7)       Lines 307-309: this is not true. What the authors demonstrated is valuable for a specific NF/SC ratio? . What if there is a different NF/SC ratio? Or rather a different crop?

Authors response:

We rephrased the sentence in the revised version. Where it was

“In the present study, it was demonstrated unequivocally that the legal width should be at least 50 meters, but preferably more, corroborating the studies of GAEMA (2012).

Now is

“In the present study, it was suggested for a range of native forest / sugar cane occupational ratios (0.7 – 3) that the legal width should be at least 45 meters, but preferably more, corroborating the studies of GAEMA [48].”

MINOR COMMENTS

Comment #1

Figure 3: please add letters (3a, 3b, 3c) for more clarity. The Legend about land use is not totally consistent with lines 158-166. For example “Floodplain” is not reported in the figures, and should be distinguished at least in sub categories such as “vegetated flood plain with threes, Vegetated flood plain with grass, non-vegetated floodplain”. Or just eliminate it including those areas in other land use classes

Authors response:

The figures were revisited according to the reviewer suggestions. The text was also adjusted to adhere more tightly to the figure legends.

Comment #2

Lines 297-305: I suggest to move it to the introduction section, where the laws are presented.

Authors response:

We acknowledge the comment. However, in this case we prefer to keep the sentence in the discussion, because the paper is focused on evaluation of a law and therefore some discussion about the law is also necessary.

Comment #3

From which reference point did the authors measure the buffer width? Widest riverbank, or ordinary? Could you (or does the law) provide a return period to detect the reference width of the riverbed?

Authors response:

We used the new law reference for the sake of comparison. No we do not have a return period for the reference width.

Reviewer 2 Report

The investigations are ok but I have number of general and detailed remarks

General remark.

- the investigation period is just one year and the is no information about rainfalls during this year, so there is no information if this year was average dry or wet. If you have just one year this information is important if you want predict something in the future.

- the sediment transport (TDS, Turb.) is strongly connected with the rainfall intensity. There is also no information when you took a water samples - during rainfall, after rainfall, during low flow period etc.

-Your IWQ is based on average. How you calculate average - is just simple average or may be flow weighted average. The way you calculate average give you different answers.

-If you take a look on your data (not calculated IWQ), water quality is not so bed. The TDS is slightly increased but the other values are good with one exception - are you sure about your conductivity units? 60S/cm = 60 000 miliS/cm. The concentrated acid has about 1000 miliS/cm. Clean river water has about 500 mikroS/cm.

Detailed remarks:

1. row 34 I do not understand  "ground truth validation"

2. It looks that the experimental sites a catchment or subchatchments (at least based of Fig2 I have such impression). Could you give the information about area of this sides? In row 181 you write sub - basin but, for example fig.3 is entitled, experimental side. It makes me confused.

3. row 205 -OD Dissolved oxygen. May be better DO?

4. Table 2. It is not clear what's b*** , a*** etc. May be it will be better to put the explanation under the table.

5. 218-219 could you explain the weights for this particular indicator. I mean why for example pH is 0,21 and Od i 0,2. Is it a standard procedure, already proved that this weight should be like this, or you just judge yourself. Why you exclude ORP from your IWQ. ?You have data but you do not use it so why you show it. 

 6. row 223. ArcMap is not fundamental. Qgis is cheaper (for free) and you can do the same thing. Please avoid such statement in scientific papers.

7. Could you show a sampling points on the figures 3. By the way, you have 3 figures  with  the same numbers.

8. Fig.4. The r2 parameter for IWQ against the buffer with is 1. It is generally the problem of polynomial function. You can always find the relation with r2 equal 1 if you use the polynominal function. This is mistake, this kind of function do not explain anything.  Did you check the statistical significance of your relations? Probably not, and it will be very hard based on 3 points.

9. row 263. I thing you do not have enough data to write something like that. You can not judge (rows264 -266) based on such a limited data. Your results is just a suggestion that maybe, you have a relation, but this is not a prove.

10. Could you explain how did you calibrated TDS for water. I guess, the equipment you use, recalculated the Turbidity into TDS, but it usually give a big mistake. In this case, the TDS is the most important because the buffers (riparian or just buffer strips) have a strong impact specially on water (rainfall) erosion and particle transport from land into waters. 

11. Row 296 But you do not included the nutrient into your IWQ.

12. row 306-307. I thing this should be more precise presented in the Introduction. Because you generally do not have much data I suggest show your results as a special case of the other research.

13 . 320-321. I do not agree that you prove that. It is not enough data. It is just a suggestion.

Author Response

The investigations are ok but I have number of general and detailed remarks

General remark #1

- The investigation period is just one year and the is no information about rainfalls during this year, so there is no information if this year was average dry or wet. If you have just one year this information is important if you want predict something in the future.

Authors’ response:

We agree with the reviewer and thank the comment. The annual rainfall in Uberaba in 2016 was 1261.81 mm, which means this year was dry because the long term average rainfall is 1584.2 mm.

To inform about the weather conditions in 2016, the following sentence was added to the revised version:

“The annual rainfall in 2016 was 1214.37mm. This value is smaller than the long-term average (1584.2 mm), meaning that 2016 was a dry year.”

General remark #2

- The sediment transport (TDS, Turb.) is strongly connected with the rainfall intensity. There is also no information when you took a water samples - during rainfall, after rainfall, during low flow period etc.

Authors’ response:

The reviewer is right. In the revised version we added a table (Table 1) where the daily rainfall is presented for the sampling day and three antecedent days. We also added a text to summarize the information depicted in the table, namely the following sentences:

“The annual rainfall in 2016 was 1214.4mm. This value is smaller than the long-term average (1584.2 mm), meaning that 2016 was a dry year. Each month, the samples were collected between calendar days 15 and 20. The weather conditions in the sampling day as well as during the three antecedent days are summarized in Table 1. In the sampling day, rainfall was always < 5 mm with the exception of February 2016 and January 2017 campaigns when rainfall reached 5.9 and 10.9 mm, respectively. In the antecedent days, average rainfall was also small (3.5 – 5.8 mm), with few exceptions represented in boldface in Table 1. The antecedent days with a substantial rainfall were November 13, 2016, and January 16, 2017, with precipitation > 25 mm. Therefore, the average analytical results should reflect long-term effects of land use and buffer strip width on the quality of stream water rather than short term effects related with storm events.”

General remark #3

-Your IWQ is based on average. How you calculate average is just simple average or may be flow weighted average. The ways you calculate average give you different answers.

Authors’ response:

This is a very pertinent question. In fact we calculated the IWQ using the CETESB equation (Eq. 1), which relies on simple arithmetic average.

General remark #4

-If you take a look on your data (not calculated IWQ), water quality is not so bed. The TDS is slightly increased but the other values are good with one exception - are you sure about your conductivity units? 60S/cm = 60 000 miliS/cm. The concentrated acid has about 1000 miliS/cm. Clean river water has about 500 mikroS/cm.

Authors’ response:

Yes, we agree. But if you look carefully to Equation 1 you will notice that the IWQ has a multiplicative formulation, which makes it rather sensitive to small changes in the bearing parameters. That is the reason why relatively good values of individual parameters result in poor quality.

In the revised version we improved this part of the manuscript by incorporation a figure to describe the standardization of quality parameters into the q scores of Equation 1, which expresses the sophistication of this index. We also made a note on the multiplicative character of Equation 1, as follows:

It is worth to note that the IWQ index is rather sensitive to small changes in the bearing parameters, given the multiplicative formulation of Equation 1. As corollary of this, a good water quality (IWQ > 51) requires that all q values are high while an excellent quality (IWQ > 79) implies that all q scores are very high.”

Detailed remarks:

1. row 34 I do not understand  "ground truth validation"

Response: we changed to “verification”. Although land uses have been compiled from existing land use maps they have been verified in the field given the detailed scale of this study.

2. It looks that the experimental sites a catchment or subchatchments (at least based of Fig2 I have such impression). Could you give the information about area of this sides? In row 181 you write sub - basin but, for example fig.3 is entitled, experimental side. It makes me confused.

Response: in fact the experimental sites are three small headwater catchments. The following modifications were made to the revised version to clarify this point. The heading of Section 2.2 changed

from

“2.2. Experimental sites”

 to

“2.2. Experimental sites (sub-catchments)

Further, in the first sentence of this section, the areas of each sub-catchment are provided:

“The experimental sites comprised three sub-basins selected within the EPA-URB, termed Mangabeira 1 (373.09 ha), Mangabeira 2 (426.6 ha) and Lanhoso (1243.64 ha) (Figure 2).”

Finally, in figures 3a-c, “experimental site” has been replaced by “sub-basin”.

3. row 205 -OD Dissolved oxygen. May be better DO?

Response: Done

4. Table 2. It is not clear what's b*** , a*** etc. May be it will be better to put the explanation under the table.

Response: Ok. The legend of this table (now Table 3), was changed to:

“Table 3 – Analytical results for the water samples: average value, standard deviation, Tukey test result. The symbols were defined in the text (Section 2.3.2). Values with different label (lowercase letters a, b, c or d) are considered significantly different from each other by the Tukey test (ANOVA) (p < 0.05), and therefore differentiate the sub-basins.”

5. 218-219 could you explain the weights for this particular indicator. I mean why for example pH is 0,21 and Od i 0,2. Is it a standard procedure, already proved that this weight should be like this, or you just judge yourself. Why you exclude ORP from your IWQ. ?You have data but you do not use it so why you show it.

Response: The weights were not decided by us. We followed the recommendations of reference nº 30, who used the IWQ with less than the 9 original variables.

RIBEIRO, I. V. A. DE S., BOUCHONNEAU, N., SILVA, A. C. DA, FERNANDES, R. M. C.; PINHEIRO, L. de S. Cálculo do índice de qualidade de água (IQA), com estudo de caso nos rios Cocó e Maranguapinho, Ceará. In: SIMPÓSIO BRASILEIRO DE RECURSOS HÍDRICOS, 18, Campo Grande, 2009.

 6. row 223. ArcMap is not fundamental. Qgis is cheaper (for free) and you can do the same thing. Please avoid such statement in scientific papers.

Response: Yes it is true that QGIS is free. But we did not use QGIS in this study. We have a Campus License for ArcMap that serves hundreds of students and professors, and in fact we used ArcMap in this study. Therefore, I believe we should refer to ArcMap because it is the truth.

7. Could you show a sampling points on the figures 3. By the way, you have 3 figures  with  the same numbers.

Response: In the revised version Figure 3 now represents the sampling sites at the outlet of the catchments, and the three panels are identified as 3a, 3b and 3c.

8. Fig.4. The r2 parameter for IWQ against the buffer with is 1. It is generally the problem of polynomial function. You can always find the relation with r2 equal 1 if you use the polynominal function. This is mistake, this kind of function do not explain anything.  Did you check the statistical significance of your relations? Probably not, and it will be very hard based on 3 points.

Response: The reviewer is right. However, in the revised version the parabolic equation is not used with any predictive purposes. It is merely indicative. Besides, the rationale behind the parabolic behavior is now justified, in response to other reviewer:

“The parabolic trend in Figure 5a may reflect the fact that water quality in MG1 is affected by water quality of MG2, besides the influence of land use and buffer strip width. Because the IWQ in both catchments are similar, regardless the differences between buffer strip width, the contribution from MG2 is probably large. Put another way, if water quality in MG1 reflected solely land use and buffer strip width, then a lower IWQ would be expected in this catchment. In the figure the MG1 point would drop along the y-axis and the general trend would shift from the parabolic trend towards a (more likely) linear trend.”

9. row 263. I thing you do not have enough data to write something like that. You can not judge (rows264 -266) based on such a limited data. Your results is just a suggestion that maybe, you have a relation, but this is not a prove.

Response: We very much appreciate this comment, also made by other reviewer. In the sentence:

“Therefore, the 30 m threshold foreseen in Federal Law n.º 12651/2012 may not satisfy the water quality requirements in the studied basins”

we changed “do not” by “may not”. Besides, following that sentence we added the next sentence to correctly bind our speech:

It should be admitted, however, that this study was based on a small number of catchments and that more general conclusions would require a more exhaustive analysis based on a larger sample. Besides, the interpretations so far hold for anthropogenic catchments used for sugar cane production and cannot be extended directly to other land uses. Finally, the weather reference for this study is a dry year.”

10. Could you explain how did you calibrated TDS for water. I guess, the equipment you use, recalculated the Turbidity into TDS, but it usually give a big mistake. In this case, the TDS is the most important because the buffers (riparian or just buffer strips) have a strong impact specially on water (rainfall) erosion and particle transport from land into waters.

Response: the probe used to measure TDS was calibrated for electric conductivity (Ec) and then a function was used to convert Ec values into TDS values. Turbidity values were measured with a turbidity meter.

11. Row 296 But you do not included the nutrient into your IWQ.

Response: Yes, but in that sentence we were talking about mitigation measures in general, which could lead to nutrient reduction in stream water, besides reduction of sediment loads.

12. row 306-307. I thing this should be more precise presented in the Introduction. Because you generally do not have much data I suggest show your results as a special case of the other research.

Response:  In the revised version the sentence was changed from

“In the present study, it was demonstrated unequivocally that the legal width should be at least 50 meters, but preferably more, corroborating the studies of GAEMA (2012).”

To

“In the present study, it was suggested for a range of native forest / sugar cane occupational ratios (0.7 – 3) that the legal width should be at least 45 meters, but preferably more, corroborating the studies of GAEMA [48].

13 . 320-321. I do not agree that you prove that. It is not enough data. It is just a suggestion.

Response:  We agree wit the reviewer. The word “proved” was replaced by “suggested”. The word prove was eliminated from the entire manuscript.

Reviewer 3 Report

General Comments:

            This manuscript evaluated the water quality effects of riparian buffer width using the field measured data. This paper is of great benefit to the forest/agricultural managers to understand the impacts of buffer zone widths on water quality. The paper could be improved by adding discussions on the effects of slope percent on buffer widths.

Specific comments:

1.) Line 109: The specific goal should be numbered 3.

2.) Figure 3 a-c should include buffer widths in the caption. Also, Figure 3 should be numbered a to c as mentioned in the text in line 158.

3.) Line 179: Define ‘deformed’.

4.) Line 178: This sentence says that sampling was taken atleast 10 m away from the stream. Can you be more specific? Were this samples taken within the buffer zone?

5.) Line 199: Clarify ‘margin’.

6.) Table 1: Add buffer widths. Were Al, Sand, and Silt significantly different in the three watersheds? This is not discussed in the results.

7.) One of the limitations of this paper is the absence of the analysis slope percent factor on buffer widths. I believe that the influence of slope factor on buffers should also be discussed somewhere in the “Discussions” section even though I know that no analysis was conducted. What was the average slope of buffer widths in the three watersheds? As I understand that the buffer widths also depends on slope factor and slope length.

Author Response

Reviewer #3

Specific comment #1

1.) Line 109: The specific goal should be numbered 3.

Authors’ response: Done

Specific comment #2

2.) Figure 3 a-c should include buffer widths in the caption. Also, Figure 3 should be numbered a to c as mentioned in the text in line 158.

Authors’ response: Done

Specific comment #3

3.) Line 179: Define ‘deformed’.

Authors’ response: The term “deformed was replaced by “disturbed”

Specific comment #4

4.) Line 178: This sentence says that sampling was taken at least 10 m away from the stream. Can you be more specific? Were this samples taken within the buffer zone?

Authors’ response: Yes the soil samples were collected within the buffer zone. We clarified that in the revised version: “…and took place in April and November of 2015 at least 10 m away from the stream within the buffer zone”.

Specific comment #5

5.) Line 199: Clarify ‘margin’.

Authors’ response: we replaced “margin” by “stream margin”.

6.) Table 1: Add buffer widths. Were Al, Sand, and Silt significantly different in the three watersheds? This is not discussed in the results.

Authors’ response: We do not have a soil characterization covering the entire catchment area, only within the buffer zones.

7.) One of the limitations of this paper is the absence of the analysis slope percent factor on buffer widths. I believe that the influence of slope factor on buffers should also be discussed somewhere in the “Discussions” section even though I know that no analysis was conducted. What was the average slope of buffer widths in the three watersheds? As I understand that the buffer widths also depends on slope factor and slope length.

Authors’ response: In fact we do not have information on that, the reason why we could not make a discussion.

Round 2

Reviewer 1 Report

The paper was signficantly improved and is now acceptable with minor revisions.

Lines 291-296 and Figure 5: the letters used to indicate the catchments are not consistent between the figure and the text (MG1 or CM1?) and with those indicated in line 266

Figure 5a: the trend you show for BW-IWQ relationship appears to grow infinitely for increasing BW. You should discuss the fact that there might be a maximum value of IWQ that you could find, which is probably the IWQ that you could find in a similar catchment but in total absence of anthropic activities and total native forest cover. Do you have any reference condition to use as a “target value” that a stream should have in terms of IWQ? This information wouold help the reader to understand what is the impact of human activities in terms of reduction of water quality.  Iguess it is somehow reported in Figure 4, but it is not clear to me how did the authors choose the reference condition for each parameter of IWQ.

Line 397: again, ground truth data is not proving it that clearly. Please change “prove” with “suggest”

I also suggest the authors to create a “conclusion” section where to place the most important sentences that are now closing the discussion section. This would help to catch the main findings of the work also by a quick read.

Author Response

Reviewer #1

General appreciation:

The paper was signficantly improved and is now acceptable with minor revisions.

Authors’ response:

We very much thank the reviewer’s positive appreciation. We did our utmost to improve the paper and are glad that the reviewer has recognized our efforts.

Minor Comment #1

Lines 291-296 and Figure 5: the letters used to indicate the catchments are not consistent between the figure and the text (MG1 or CM1?) and with those indicated in line 266

Authors’ response

The terms MG1 and MG2 were replaced by terms CM1 and CM2, respectively.

Minor Comment #2

Figure 5a: the trend you show for BW-IWQ relationship appears to grow infinitely for increasing BW. You should discuss the fact that there might be a maximum value of IWQ that you could find, which is probably the IWQ that you could find in a similar catchment but in total absence of anthropic activities and total native forest cover. Do you have any reference condition to use as a “target value” that a stream should have in terms of IWQ? This information wouold help the reader to understand what is the impact of human activities in terms of reduction of water quality.  Iguess it is somehow reported in Figure 4, but it is not clear to me how did the authors choose the reference condition for each parameter of IWQ.

Authors’ response

We believe the answer is given in Table 5. In that table we show that even if we could duplicate the native forest we would achieve a regular water quality if the buffer strip with were wider than 45 meters. This is enough indication for the reader that the present situation should be altered.

Minor Comment #3

Line 397: again, ground truth data is not proving it that clearly. Please change “prove” with “suggest”

Authors’ response

The words “ground truth data prove” were replaced by “field data suggest”.

Minor Comment #4

I also suggest the authors to create a “conclusion” section where to place the most important sentences that are now closing the discussion section. This would help to catch the main findings of the work also by a quick read.

Authors’ response

A “Conclusions” section was added to the revised version, as suggested

Reviewer 2 Report

Dear Authors

Thank you for your answers and explanations as well as changes in the text. I think that the paper can be published in this form.

Author Response

We very much thank the appreciation made by the reviewer